# Domestic Risk Factors for Atopic and Non-Atopic Asthma in First Nations Children Living in Saskatchewan, Canada

**DOI:** 10.3390/children7050038

**Published:** 2020-04-27

**Authors:** Donna C. Rennie, Chandima P. Karunanayake, Josh A. Lawson, Shelley Kirychuk, Kathleen McMullin, Sylvia Abonyi, Jeremy Seeseequasis, Judith MacDonald, James A. Dosman, Punam Pahwa

**Affiliations:** 1College of Nursing, University of Saskatchewan, 107 Wiggins Road, Saskatoon, SK S7N 2Z4, Canada; 2Canadian Centre for Health and Safety in Agriculture, University of Saskatchewan, 104 Clinic Place, Saskatoon, SK S7N 2Z4, Canada; james.dosman@usask.ca (J.A.D.); pup165@mail.usask.ca (P.P.); 3Department of Medicine, College of Medicine, University of Saskatchewan, 103 Hospital Drive, Saskatoon, SK S7N 0X8, Canada; jal226@mail.usask.ca (J.A.L.); shelley.kirychuk@usask.ca (S.K.); 4Department of Community Health and Epidemiology, College of Medicine, University of Saskatchewan, 104 Clinic Place, Saskatoon, SK S7N 2Z4, Canada; kathleen.mcmullin@usask.ca (K.M.); sya277@mail.usask.ca (S.A.); 5Willow Cree Health Centre, Beardy’s and Okemasis First Nation, P.O. Box 96, Duck Lake, SK S0K 1J0, Canada; jseeseequasis@beardysband.com; 6William Charles Health Centre, Montreal Lake Cree Nation, P.O. Box 240, Montreal Lake, SK S0J 1Y0, Canada; judithlynmacdonald@hotmail.com

**Keywords:** Atopic asthma, non-atopic asthma, domestic environments, damp housing, Aboriginal

## Abstract

Both allergic and non-allergic asthma phenotypes are thought to vary by specific housing and other indoor environmental conditions. This study evaluated risk factors for allergic asthma phenotypes in First Nation children, an understudied Canadian population with recognized increased respiratory morbidity. We conducted a cross-sectional survey with a clinical component to assess the respiratory health of 351 school-age children living on two rural reserve communities. Asthma was defined as parental report of physician diagnosed asthma or a report of wheeze in the past 12 months. Atopy was determined by a ≥ 3-mm wheal response to any of six respiratory allergens upon skin prick testing (SPT). Important domestic and personal characteristics evaluated included damp housing conditions, household heating, respiratory infections and passive smoking exposure. Asthma and atopy prevalence were 17.4% and 17.1%, respectively. Of those with asthma, 21.1% were atopic. We performed multivariate multinomial logistic regression modelling with three outcomes: non-atopic asthma, atopic asthma and no asthma for 280 children who underwent SPT. After adjusting for potential confounders, children with atopic asthma were more likely to be obese and to live in homes with either damage due to dampness (*p* < 0.05) or signs of mildew/mold (*p* = 0.06). Both natural gas home heating and a history of respiratory related infections were associated with non-atopic asthma (*p* < 0.01). Domestic risk factors for asthma appear to vary by atopic status in First Nations children. Determining asthma phenotypes could be useful in environmental management of asthma in this population.

## 1. Introduction

Aboriginal peoples, including First Nations, Inuit and Metis groups, represent 2.6% of the Canadian population and, of these, 60.8% are First Nation [1]. Unlike their non-Aboriginal counterparts, Aboriginal children make up a disproportionally larger percentage of their population (29.0% versus 16.5%) for non-Aboriginal children [1]. Respiratory disease, including asthma, is a major cause of emergency room visits and hospitalization in First Nations and other Aboriginal children [2,3]. Although the burden of asthma is higher in Aboriginal child populations compared to same age and sex non-Aboriginal populations, the prevalence of asthma (9.5 to 14.6%) is similar [4,5,6], with household factors playing a role in disease manifestation [4,7,8]. While respiratory disease, including asthma, is a major cause of illness for Canadian Aboriginal children [7], there is limited understanding of the role of atopy in asthma prevalence [6,9] within this population.

Major environmental factors known to be associated with childhood asthma can be allergenic or non-allergenic in origin [10]. The allergic form of asthma is triggered by contact with specific environmental aeroallergens resulting in an adaptive T helper cell (TH2 high) immune response with increased eosinophilia. Conversely, the non-allergic form is characterized by a low TH2 (TH2low) adaptive immunity, primarily in response to certain microbial agents including viruses, with resulting neutrophilia [10,11]. While a history of parental atopic disease is associated with both non-atopic and atopic childhood phenotypes of asthma [12], environmental exposures seem to vary by asthma allergic phenotype [12,13].

Two recent meta-analyses concluded that there was approximately a 30–50% increased risk of childhood asthma with reported damp housing [14,15]. Dampness, moisture and mold are prevailing conditions in the homes located on the rural First Nation reserves [16,17,18]. National studies examining the health of children on reserves consistently report that homes requiring major repairs are associated with respiratory outcomes including asthma, although the potential contributing features of these homes, such as dampness and mold, have not been explored [4,6]. To date, there is limited understanding of how dampness and other housing conditions of First Nation children’s homes may contribute to increased asthma and specifically, to either non-atopic and atopic asthma. The purpose of this investigation was to assess the importance of particular housing conditions for asthma risk among First Nations children living on two rural reserve communities, with specific evaluation by non-atopic and atopic status. 

## 2. Materials and Methods

The First Nations Lung Health Study is a longitudinal study of the respiratory health of persons six years and older living in either a Plains Cree or Woodland Cree First Nation community located in rural Saskatchewan, Canada. This analysis evaluates data from the baseline cross-sectional study conducted in 2013 for children 6–17 years of age. Ethical approval for the children’s study was received from the University of Saskatchewan Biomedical Ethics Review Board (Bio Certificate #13-27). 

The detailed research protocol used for the children’s study has been previously reported [19]. School officials approved the study and surveys. We distributed questionnaires to parents through community schools and a neighboring school of one community where the enrollment primarily consisted of children from the study community. Parental consent and child assent forms for clinical testing accompanied the questionnaires. Clinical assessments conducted at the schools included anthropometric measurements and skin prick testing (SPT) to local aeroallergens. Parents returned questionnaires and consent/assents to the school for collection by the research team. A $5.00 (CDN) incentive was provided to those returning questionnaires.

### 2.1. Asthma Phenotypes

Asthma was defined by questionnaire as either a report of physician diagnosed asthma or wheeze in the past 12 months. Atopy was assessed by SPT to six common standardized inhalant allergens: combined *d farina* and *d pteronyssinus* (house dust mite mixed, (HDM mixed)), Cladosporium, Alternaria, Aspergillus, *felis d* (cat), and mixed grasses (ALK-Abello Pharmaceuticals, Inc, Mississauga ON). In addition, negative (saline) and positive (histamine) controls were used. Atopic was defined as a positive SPT prick test to any allergen with a wheal of ≥3 mm compared to the negative control [20]. Notably, four of the six allergens chosen (HDM, cat, grass, and Alternaria alternata) are known to account for approximately 94% of atopy in young children [21]. The asthma phenotype was defined by atopic status and was categorized into three groups: atopic asthma, non-atopic asthma, and no asthma.

### 2.2. Moisture/Mold and Other Housing Conditions

We identified damage caused by dampness (damp damage) by the question: “Does your home have any damage caused by dampness (e.g., wet spots on walls, floors)? (yes/no)”. Mold/mildew was determined by the question: “Are there signs of mold or mildew in any living areas of your home? (yes/no)” and home dampness by a question that identified water or dampness in the home in the past 12 months due to broken pipes, leaks, heavy rain or flooding (yes/no). Other dichotomized (yes/no) variables included having a cat or dog in the home and environmental tobacco smoke (ETS) exposure (child smoked or smoking occurred in the home). A crowding index was calculated as a ratio of number of individuals in the home to the number of rooms excluding bathrooms and hallways. We also assessed use (yes/no) of a dehumidifier, humidifier, air conditioner, air filter in the home and the type of home heating (natural gas or other).

### 2.3. Personal Characteristics

Body mass index (BMI) was calculated from measured height and weight using the standard BMI formula (weight (Kg)/height (cm)^2^). Children were then classified as obese, overweight and normal/underweight according to age and sex-standardized BMI [22]. Height was measured against a wall in centimeters (cm) using a fixed tape measure. Weight was measured in kilograms (Kg) using a spring scale. Children performed both procedures wearing indoor clothing and stockinged feet.

Other variables assessed by questionnaire included parental education (<Grade 12, ≥Grade 12) and a parental history any of the following atopic conditions: asthma, allergy, eczema or hay fever. Respiratory-related infections (RRI) were assessed as a history of any of the following: croup, bronchitis, pneumonia, tonsillitis, ear infection and/or the use of antibiotics in the past 12 months to treat a respiratory problem (Yes/No).

### 2.4. Statistical Analysis

Data were analyzed for the 280 children who underwent SPT. Univariate associations between the three outcome groups (atopic asthma, non-atopic asthma, and no asthma) and household and personal characteristics were assessed by χ^2^ test for proportions and one-way analysis of variance for continuous data. There were moderately significant correlations (*p* < 0.05) observed between damp damage, damp housing, visible mold and mildew. Because of this and the strong conceptual relationships between the three variables, we analyzed each of these dampness indicators separately. Multinomial logistic regression analyses were conducted to assess relationships between different measures of dampness and mold and asthma phenotypes. The strength of associations were assessed by the odds ratios (OR) and 95% confidence intervals (CI). Potential effect modifiers significant at the *p* < 0.10 level or those known to be frequently associated with asthma in previous studies were included. We used SPSS Version 24 (IBM SPSS Statistics for Windows. Armonk, NY, USA: IBM Corp., 2017) for all analyses.

## 3. Results

There were 351 school-age children who completed the survey for a response rate of 57.7%. The mean age of students was 10.9 ± 3.1 years and more girls (55.0%) than boys participated in the study. Asthma prevalence did not differ between the two communities and was significantly higher in boys (21.8%) than in girls (13.4%, *p* < 0.05). In total, there were 47 children (13.4%) with a report of wheeze in the past 12 months and 61 (17.4%) children who had a diagnosis of asthma only. Of those with asthma, 36.1% reported wheezing in the past 12 months.

We found no differences in demographic factors or housing characteristics between those who underwent SPT and those who did not (data not presented). Of the 280 children who underwent SPT, 48 (17.1%) were atopic. Of those with a positive SPT, 23 (47.9%) were sensitized to HDM mixed and 19 (39.6%) to cat dander. The proportion of children sensitized to any mold was 12.5%. Of those undergoing SPT, 71 (25.4%) had asthma. Atopy was identified in 15 (21.1%) children with asthma and in 33 (15.8%) children with no asthma. Atopy and asthma status did not differ by community.

Table 1 presents the personal and housing characteristics of children by asthma phenotype. Within asthma groups, the prevalence of obesity was almost twice as high in the atopic asthma group compared to the non-atopic asthma group and more than double of that observed in the non-asthma group. Reported parental atopy was similar in the atopic and non-atopic asthma groups and higher than what we observed in children with no asthma. There were 58% of the children with a history of RRIs and the prevalence of such infections was highest in the non-atopic asthma group (*p* < 0.001).

Household dampness varied from 40.4% for damage due to dampness to 54.3% for recent evidence of water or leaks in the home (Table 1). Of the three outcome groups, children with atopic asthma were most likely to live in homes with damp damage or in those with signs of mold/mildew. The use of air quality measures in homes was infrequent and did not differ by group. While natural gas use was the most common form of heating, its use was highest in homes of children with non-atopic asthma (*p* < 0.01). ETS exposure was prevalent with 55% of children exposed. This was higher in the non-atopic asthma group. Notably, 88% of children had at least one parent who smoked. Table 2 presents the univariate odds ratios and 95% CI for important household and personal factors considered in the multivariate analyses.

The results of the adjusted multinomial logistic regression models for reports of dampness or mold in the home by atopic status are presented in Table 3 (Models 1, 2, 3). Compared to children with no asthma and consistent with the findings from the univariate regression, children with atopic asthma were significantly more likely to live in homes with damage due to dampness (Model 1). As shown in Model 2, there continued to be a borderline statistical association between a report of visible mold or mildew in the home and atopic asthma (*p* = 0.06). A report of a home dampness event in the past 12 months was not associated with atopic or non-atopic asthma. While obesity was associated with an increased risk of atopic asthma in all models, natural gas heating and a history of an RRI were consistently associated with non-atopic asthma in all models.

## 4. Discussion

In this study, reports of both damage to the home caused by dampness and the presence of visible mold or mildew in the home were associated with atopic asthma but not with non-atopic asthma. There were clear differences in associations by phenotype for other risk factors as well. While obesity was associated with atopic asthma, natural gas heating in the home and RRIs were associated with non-atopic asthma.

Asthma prevalence in this study at 17.4% (with 95% CI: 13.4–21.3%) is similar to that reported regionally in rural, non-Aboriginal child populations (14.7–18.9%) [13,19,23,24]. While atopy measured by SPT at 17.8% was comparatively lower (19.4–22.4%) [24,25]. Our data show that asthma prevalence appears to be higher than what has been previously reported in other Canadian First Nation child populations. Using data from the 2003 Regional First Nations Survey, Senthilselvan et al. [6] reported asthma in Canadian First Nations children five to 11 years old living on reserves to be 15.6%. Chang et al. noted an asthma prevalence of 14.3% in off-reserve First Nations children aged 6–14 years participating in the 2006 Aboriginal Peoples Survey [8]. A comparable asthma prevalence of approximately 18% was observed for American Indian/Alaskan Native children participating in the National Health Interview Survey 2001–2005 [26]. The increase in asthma prevalence in the current study is modest and may be due to use of different asthma definitions and the variability of assessment strategies used between studies. However, the generally poor housing conditions observed in the reserve communities and the high prevalence of smoking found in homes (52.7% of all study homes) could also contribute to higher asthma prevalence seen in the current study.

While the burden associated with asthma was not a primary focus of this study, two potential measures of health burden within the study population including school absenteeism for three days or more due to chest illness and previous hospitalization for a chest illness were examined retrospectively and it appears that children with asthma have increased morbidity. When compared to children without asthma, children with asthma were proportionally more likely to be absent from school (*p* < 0.001) because of a chest illness or to require hospitalization for a chest illness (*p* < 0.001). Studies describing the health burden associated with asthma in First Nations population are limited [4,8] but report similar findings for healthcare usage.

Our findings for damp damage and visible mold support certain earlier findings by others [27,28]. Children participating in the 2005–2006 National Health and Nutrition Survey (NHANES III) were at increased risk for atopic asthma (total IgE > 170 KU/L) with a report of mildew odor in the home [27]. Su identified higher total and specific IgE levels in children with elevated fungal counts in their homes [28]. Evidence of positive associations between atopic asthma and damp housing are limited with a recent systematic review demonstrating that damp housing was more likely to be associated with non-atopic asthma rather than atopic asthma [12]. The inconsistencies in findings from studies could be partially due to the classification of dampness in studies or it could reflect the dynamic nature and problem of both the allergic and non-allergic agents of mold occurring simultaneously in damp housing environments [29]. Mycotoxin, volatile organic compounds (VOC’s) or microbial agents such as endotoxin have been observed in moldy environments and could contribute to a non-atopic inflammatory response in children with asthma [30], whereas the proliferation of spores from mold species common to moldy domestic environments could produce an allergenic inflammatory response in children [27]. Objective evaluation of the broad array of microbial agents in damp homes at the time of study and over time would help to identify the specific microbial characteristics involved in the pathogenesis of atopic asthma phenotypes. While the kinds of microbial agents found in households may be important for developing immune responses, so might their concentration at certain critical times in the children’s lives [10].

Over 40% of homes in the study had damage due to dampness. This is higher than that reported in recent regional cross-sectional studies with non-Aboriginal children using the same definitions of dampness (20.5%) [23] and is very similar to findings from previous studies evaluating mold and dampness in First Nations reserve homes (44 to 54%) [17,18]. Moisture continues to be a significant environmental problem in reserve homes and, as the findings from this study demonstrate, it may be a particular problem for those children with atopic asthma who live in such communities.

The associations we found between previous RRIs and non-atopic asthma support the findings from a recent review of predominant risk factors for non-atopic asthma [12]. Infectious respiratory incidences appear important for development of a specific allergic asthma phenotype. Recent evidence suggests that the relationship between respiratory infections and the occurrence of atopic or non-atopic asthma could be time specific where very early, severe respiratory infections can trigger asthma in atopic children, whereas chronic, low-level respiratory infections are more likely associated with development of non-atopic asthma [31].

In this study, natural gas as the main heating source was significantly higher in homes of children with non-atopic asthma. Similarly, NO_2_ levels in homes, many with gas stoves, was associated with non-atopic asthma in children participating in the NHANES III study [32]. The evidence for associations between natural gas heating or other indoor air pollutants and non-atopic asthma is preliminary and further studies should objectively measure indoor air quality and the relationship with asthma phenotypes.

### Limitations

The cross-sectional nature of this study implies associative and non-causal relationships between asthma phenotypes and variables assessed. Further study to confirm causal associations are required. We used a questionnaire self-report of mold and dampness that may not accurately reflect actual damp housing conditions of homes [33]. Furthermore, it is possible that those who were more concerned about mold or water damage in their homes and who had a child with asthma or wheeze would be more likely to report its presence. Home owners have been shown to underestimate mold exposures in the home if they smoke or to over report household fungi exposure if they have allergies [33]. While the questions used to evaluate dampness and mold in the home were found to be consistent with unbiased housing inspections by others [34], objective characterization of the indoor environment for mold species, mycotoxins and endotoxin related biomarkers is still warranted.

We cannot exclude potential misclassification of asthma cases in the study, although a parental report of doctor-diagnosed asthma in questionnaires shows good correlation with observed physician-evaluated asthma [35]. We included a history of current wheeze in our definition, which could have misclassified some cases of wheeze as asthma. However, the usefulness of including current wheeze in the past 12 months as an indicator of asthma has been previously identified [36]. While atopy was based on objective measurement of allergy by SPT, asthma was based on parental report by questionnaire. To assess if there was misclassification of our definition of asthma that included wheeze in the past 12 months, we conducted a post hoc sensitivity analysis with the multinomial logistic regression models, with just children in the who had a report of doctor diagnosed asthma with or without atopy. The findings from the sensitivity analysis were consistent with the original analyses for all variables except for natural gas in the home, which was no longer associated with non-atopic asthma. We therefore conclude our main findings are fairly robust, excluding findings for natural gas heating which may be more likely associated with wheeze.

Other potential limitations of this study include the lower response rate to the study and the analysis of domestic and personal risk factors confined to those who consented to SPT for atopy (*n* = 280). According to the 2006 Canadian Census for these communities, younger children (age 5–9 years) in the study were proportionally represented while middle school children (10–14 years) were overrepresented and high school children (15–19 years) were underrepresented. Females were overrepresented by 5% in the study as well. Findings in this study should be viewed cautiously for older children and by sex [37]. When we compared those who participated in the screening for atopy (*n* = 280) with those who did not (*n* = 71), there were no significant differences between groups on the variables reported in Table 1 except for RRIs, whereby more children reporting a history of respiratory infections were more likely to participate in screening for atopy.

We found no association between ETS exposure and asthma phenotypes, although ETS exposures evaluated in other studies do show associations with non-atopic asthma [12]. Non-significant findings between ETS and respiratory outcomes have been recorded in studies conducted in Aboriginal reserve communities [6,38]. The endemic nature of smoking in the studied communities may account for the non-significant findings for ETS and asthma. Prospective evaluation of the long-term effects of pervasive smoking exposures and lung health of Aboriginal children is still required.

## 5. Conclusions

In conclusion, we have demonstrated important associations between atopic asthma and damp and moldy domestic environments of First Nations children. As many of the children with asthma were non-atopic, objective evaluation of microbial contaminants and ventilation within these homes and their association with specific asthma phenotypes could assist in targeted asthma management and better environmental control.

## Figures and Tables

**Table 1 children-07-00038-t001:** Personal and housing characteristics by asthma phenotype for 280 First Nation children.

Variable	Total *n* = 280	Atopic Asthma *n* = 15	Non-Atopic Asthma *n* = 56	No Asthma *n* = 209	*p* Value ^#^
	Mean ± SD	Mean ± SD	Mean ± SD	Mean ± SD	
Age in years	10.9 ± 3.1	12.4 ± 3.1	11.0 ± 3.0	10.8 ± 3.1	0.13
Crowding index	1.6 ± 0.7	1.5 ± 0.5	1.6 ± 0.6	1.6 ± 0.7	0.76
	*n* (%)	*n* (%)	*n* (%)	*n* (%)	
Sex					
Male	126 (45.0)	10 (66.7)	30 (53.6)	86 (41.1)	0.06
Female	154 (55.0)	5 (33.7)	26 (46.4)	123 (58.9)	
Body mass index					
Obese	61 (22.0)	7 (46.7)	14 (25.0)	40 (19.4)	0.12
Overweight	87 (31.4)	4 (26.7)	19 (33.9)	64 (31.1)	
Normal	129 (46.6)	4 (26.7)	23 (41.1)	102 (49.5)	
Respiratory related infections					
Yes	165 (58.9)	8 (53.3)	46 (82.1)	111 (53.1)	<0.001
No	115 (41.1)	7 (46.7)	10 (17.9)	98 (46.9)	
Passive smoking					
Yes	154 (55.0)	10 (66.7)	36 (64.3)	108 (51.7)	0.16
No	126 (45.0)	5 (33.3)	20 (35.7)	101 (48.3)	
Parental history of allergic conditions					
Yes	112 (40.0)	8 (53.3)	29 (51.8)	75 (35.9)	0.05
No	168 (60.0)	7 (46.7)	27 (48.2)	134 (64.1)	
Parental education					
<Grade 12	146 (52.1)	5 (33.3)	26 (46.4)	115 (55.0)	0.17
≥Grade 12	134 (47.9)	10 (66.7)	30 (53.6)	94 (45.0)	
Fuel type					
Natural gas	229 (81.8)	13 (86.7)	52 (92.9)	164 (78.5)	0.04
Other	51 (18.2)	2 (13.3)	4 (7.1)	45 (21.5)	
Air conditioner in home					
Yes	90 (32.1)	5 (33.3)	18 (32.1)	67 (32.1)	0.99
No	190 (67.9)	10 (66.7)	38 (67.9)	142 (67.9)	
Air filter in home					
Yes	120 (42.9)	3 (20.0)	20 (35.7)	97 (46.4)	0.07
No	160 (57.1)	12 (80.0)	36 (64.3)	112 (53.6)	
Humidifier in home					
Yes	37 (13.2)	1 (6.7)	10 (17.9)	26 (12.4)	0.42
No	243 (86.3)	14 (93.3)	46 (82.1)	183 (87.6)	
Dehumidifier in home					
Yes	25 (8.9)	4 (26.7)	5 (8.9)	16 (7.7)	0.05
No	255 (91.1)	11 (73.3)	51 (91.1)	193 (92.3)	
Wood fireplace in home					
Yes	8 (2.9)	1 (6.7)	1 (1.8)	6 (2.9)	0.60
No	272 (97.1)	14 (93.3)	55 (98.2)	203 (97.1)	
Pet cat/dog in home					
Yes	137 (48.9)	7 (46.7)	28 (50.0)	102 (48.8)	0.97
No	143 (51.1)	8 (53.3)	28 (50.0)	107 (51.2)	
Damage caused by dampness					
Yes	113 (40.4)	11 (73.3)	25 (44.6)	77 (36.8)	0.02
No	167 (59.6)	4 (26.7)	31 (55.4)	132 (63.2)	
Signs of mold or mildew in home					
Yes	121 (43.2)	11 (73.3)	23 (41.1)	87 (41.6)	0.05
No	159 (56.8)	4 (26.7)	33 (58.9)	122 (58.4)	
Home dampness past 12 months					
Yes	152 (54.3)	11 (73.3)	34 (60.7)	107 (51.2)	0.14
No	128 (45.7)	4 (26.7)	22 (39.3)	102 (48.8)	

^#^ For the categorical variables, chi squared test for proportions *p* value reported and for the continuous variables, F test *p* value reported.

**Table 2 children-07-00038-t002:** Univariate multinomial logistic regression analyses (odds ratio (OR) and 95% confidence intervals (CI)) of variables added to the multivariate analyses assessing risk factors for atopic and non-atopic asthma (referent: no asthma) for 280 First Nation children.

Variable (Referent)	Atopic Asthma *n* = 15 OR_unadjusted_ (95% CI)	Non-Atopic Asthma *n* = 56 OR_unadjusted_ (95% CI)
Age, in years	1.2 (1.00, 1.41) *	1.0 (0.93, 1.13)
Crowding index		
Sex (Female)	2.9 (0.94, 8.66)	1.7 (0.91, 2.99)
Body Mass index (Normal/underweight)		
Obese	4.5 (1.24, 16.08) *	1.6 (0.73, 3.31)
Overweight	1.6 (0.38, 6.60)	1.3 (0.66, 2.61)
Respiratory related infections (No)	1.0 (0.35, 2.88)	4.1 (1.95, 8.48) **
Passive smoking (No)	1.9 (0.62, 5.66)	1.7 (0.91, 3.09)
Parental education (≥Grade 12)		
Less than grade 12	0.4 (0.13, 1.24)	0.7 (0.39, 1.28)
Fuel type (Other)		
Natural gas	1.8 (0.39, 8.19)	3.6 (1.22, 10.39) *
Damage caused by dampness (No)	4.7 (1.45, 15.32) *	1.4 (0.76, 2.51)
Signs of mold or mildew in home (No)	3.9 (1.19, 12.51) *	1.0 (0.54, 1.78)
Home dampness past 12 months (No)	2.6 (0.81, 8.50)	1.5 (0.81, 2.69)

* *p* < 0.05; ** *p* < 0.01.

**Table 3 children-07-00038-t003:** Multivariate multinomial logistic regression model results (odds ratio (OR) and 95% confidence intervals (CI)) assessing damp housing characteristics for atopic and non-atopic asthma (reference to no asthma).

Variable (Referent)	Atopic Asthma OR_adjusted_ (95% CI)	Non-Atopic Asthma OR_adjusted_ (95% CI)
**Model 1 ^†^ (Damage Caused by Dampness)**		
Body Mass Index (Normal/underweight)		
Obese	5.9 (1.31, 26.34) *	1.6 (0.67, 3.65)
Overweight	1.5 (0.32, 6.96)	1.1 (0.51, 2.22)
Ever had infections (No)	0.6 (0.17, 2.02)	3.6 (1.65, 7.65) **
Passive smoking (No)	1.7 (0.38, 5.69)	1.8 (0.90, 3.68) ***
Fuel type (Other/combine fuel)		
Natural gas	2.8 (0.55, 14.38)	4.5 (1.44, 14.06) *
Damage caused by dampness (No)	5.5 (1.43, 20.99) *	1.2 (0.60, 2.38)
**Model 2 ^†^ (Signs of Mold or Mildew)**		
Body Mass Index (Normal/underweight)		
Obese	6.8 (1.54, 29.91) *	1.6 (0.69, 3.75)
Overweight	1.6 (0.36, 7.42)	1.1 (0.54, 2.32)
Ever had infections (No)	0.7 (0.19, 2.15)	3.7 (1.73, 8.03) **
Passive smoking (No)	1.7 (0.47, 6.38)	2.0 (0.98, 3.89) ***
Fuel type (Other)		
Natural gas	2.6 (0.50, 13.53)	4.3 (1.37, 13.26) **
Signs of mold or mildew in home (No)	3.6 (0.99, 12.91) ***	0.8 (0.41, 1.55)
**Model 3 ^†^ (Home Dampness Past 12 Months)**		
Body Mass Index (Normal/underweight)		
Obese	8.9 (2.00, 39.11) **	1.6 (0.70, 3.71)
Overweight	1.9 (0.41, 8.46)	1.1 (0.51, 2.22)
Ever had infections (No)	0.6 (0.19, 2.14)	3.5 (1.63, 7.53) **
Passive smoking (No)	1.9 (0.50, 6.84)	1.8 (0.88, 3.56)
Fuel type (Other)		
Natural gas	3.0 (0.59, 14.78)	4.6 (1.48, 14.46) **
Home dampness past 12 months (No)	3.4 (0.92, 12.89) ***	1.5 (0.75, 2.81)

* *p* < 0.05; ** *p* < 0.01; *** borderline (*p* < 0.10); ^†^ adjusted for age, sex, crowding index and parental education.

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
