# Peer review of "Domestic Risk Factors for Atopic and Non-Atopic Asthma in First Nations Children Living in Saskatchewan, Canada"

_children, 2020, doi:10.3390/children7050038_

Round 1

Reviewer 1 Report

In this paper, the authors describe associations between risk factors of the home environment and self-reported asthma prevalence in a first nation population. Their findings point towards the elevated prevalence of dampness and mold in the home conditions, which shows associations with asthma prevalence and may provide an incentive to improve housing conditions.   However, I have a series of comments that I would like the authors to address.  

  1. the authors state that previous studies have found similar prevalence rates, but increased burden of asthma in first nation populations. in their current study, they report on dampness and mold being prevalent in this sample, and show associations of dampness and mold with asthma diagnosis (atopic asthma). in light of this, it would be interesting to (a) include a statement in the discussion about the prevalence of asthma in this sample, compared to other first nations/general north american populations. (b) discuss reasons how their findings square with previously-reported similarities in asthma diagnoses across first nation and other north american samples. (c) discuss the absence of a measure of asthma burden (potentially discussing the use of reported wheeze as a measure of not-controlled asthma or increased burden instead of "asthma").
  2. the authors used 2 types of definitions for asthma diagnosis in their dataset. (self-reported physician diagnosis and self reported wheeze). this may lead to heterogeneity in terms of burden or risk factors. Was there a large overlap between these two diagnoses. Are the results sensitive to including definitions of asthma based solely on self-reported physician diagnosis and self reported wheeze?
  3. previous associations between atopy and moldy/damp housing conditions have been inconsistent. Are the findings different when considering atopic/non-atopic asthma as a single group?
  4. The authors report on the limitations of using self-report for documenting housing conditions. It would be interesting to add to this an additional problem with the use of self report with cross-sectional data; i.e. that individuals with asthma may be more bothered by their housing conditions (perceiving them to be more likely to be causing them harm, and therefore more aware of their housing conditions).

Reviewer 2 Report

The authors use the term “First Nations” in their title but study population is two specific groups of Cree children.  The individual group names are not necessarily familiar names outside Canada.  In the United States, there are many culturally diverse groups of Native American that reside in markedly different microenvironments.  If environmental dampness is important, will be important to understand in what parts of the country these people live.  I wonder if it might be better to make the title and intro more targeted on the group that they studied.

Line 134: Are there any differences in baseline demographic characteristics between the whole school population and the 57% who completed the survey?  In other words, from a demographic standpoint, is the population surveyed representative of the whole group?

In line 206, the authors state the risk of dampness is 15% higher in the study population than non-aboriginal. Is this absolute risk or relative risk?

Round 2

Reviewer 1 Report

Thank you for these clarifications and changes to the manuscript. I have no further comments.
